# Characterization of the First Secreted Sorting Nexin Identified in the *Leishmania* Protists

**DOI:** 10.3390/ijms25074095

**Published:** 2024-04-07

**Authors:** Olympia Tziouvara, Marina Petsana, Drosos Kourounis, Amalia Papadaki, Efthimia Basdra, Georgia G. Braliou, Haralabia Boleti

**Affiliations:** 1Intracellular Parasitism Group, Department of Microbiology, Hellenic Pasteur Institute, 11521 Athens, Greece; oltziouvara@pasteur.gr (O.T.); marina.kpetsana@gmail.com (M.P.); dkourounis@pasteur.gr (D.K.); apapadaki@eof.gr (A.P.); 2Department of Biological Chemistry, Medical School, National and Kapodistrian University of Athens, 11527 Athens, Greece; ebasdra@med.uoa.gr; 3Department of Computer Science and Biomedical Informatics, University of Thessaly, 2–4 Papasiopoulou Str., 35131 Lamia, Greece; gbraliou@dib.uth.gr; 4Bioimaging Unit, Hellenic Pasteur Institute, 11521 Athens, Greece

**Keywords:** sorting nexin, PX domain, BAR domain, phosphoinositide binding protein, SNX-BAR, SNX2, *Leishmania donovani*

## Abstract

Proteins of the sorting nexin (SNX) family present a modular structural architecture with a phox homology (PX) phosphoinositide (PI)-binding domain and additional PX structural domains, conferring to them a wide variety of vital eukaryotic cell’s functions, from signal transduction to membrane deformation and cargo binding. Although SNXs are well studied in human and yeasts, they are poorly investigated in protists. Herein, is presented the characterization of the first SNX identified in *Leishmania* protozoan parasites encoded by the *Ld*BPK_352470 gene. In silico secondary and tertiary structure prediction revealed a PX domain on the N-terminal half and a Bin/amphiphysin/Rvs (BAR) domain on the C-terminal half of this protein, with these features classifying it in the SNX-BAR subfamily of SNXs. We named the *Ld*BPK_352470.1 gene product *Ld*SNXi, as it is the first SNX identified in *Leishmania* (*L.*) *donovani.* Its expression was confirmed in *L. donovani* promastigotes under different cell cycle phases, and it was shown to be secreted in the extracellular medium. Using an in vitro lipid binding assay, it was demonstrated that recombinant (r) *Ld*SNXi (rGST-*Ld*SNXi) tagged with glutathione-S-transferase (GST) binds to the PtdIns3*P* and PtdIns4*P* PIs. Using a specific a-*Ld*SNXi antibody and immunofluorescence confocal microscopy, the intracellular localization of endogenous *Ld*SNXi was analyzed in *L. donovani* promastigotes and axenic amastigotes. Additionally, r*Ld*SNXi tagged with enhanced green fluorescent protein (r*Ld*SNXi-EGFP) was heterologously expressed in transfected HeLa cells and its localization was examined. All observed localizations suggest functions compatible with the postulated SNX identity of *Ld*SNXi. Sequence, structure, and evolutionary analysis revealed high homology between *Ld*SNXi and the human SNX2, while the investigation of protein–protein interactions based on STRING (v.11.5) predicted putative molecular partners of *Ld*SNXi in *Leishmania*.

## 1. Introduction

Sorting nexins (SNX), a large group of diverse proteins evolutionarily conserved in eukaryotes, are involved in many aspects of protein sorting and intracellular trafficking. SNX members are classified into different subfamilies based on the structural arrangements of their scaffolding, enzymatic, and regulatory domains [1]. They all share a common phox homology (PX) domain that binds to phosphoinositide (PI) phospholipids on the cytoplasm-facing leaflet of the plasma membrane bilayer or on intracellular organelle membranes [2,3]. This PI binding contributes to their association with cellular membranes, despite their hydrophilic nature. Many SNX family members also contain various other conserved structural domains, with Bin/amphiphysin/Rvs domains (BAR) [4] and FERMs (F for 4.1 protein, E for ezrin, R for radixin, and M for moesin [5]) being the most prevalent [2]. This modularity confers a wide variety of functions, from signal transduction to membrane deformation and cargo binding. Importantly, SNXs are crucial modulators of endosome dynamics and autophagy (discussed in recent reviews [2,6]).

Although PX domains show little sequence conservation across the SNX family members, they all possess the same core 3D structure, consisting of three antiparallel β-strands (β1–β3) followed by three α-helices (α1–α3) [7,8]. Crystal structure analysis has shown that these helices form the required scaffold for binding to the phosphoinositide (PtdIns, PIs) phosphatidylinositol-3-monophosphate (PtdIns3*P*) [9]. It is generally accepted that the PX domains of SNX family members bind predominantly to PtdIns3*P*, a signaling lipid enriched in the early endosome membrane [10,11]. Recent evidence, however, indicates that SNX proteins can also bind to other PtdIns phospholipids such as PtdIns(3,4)*P*_2_, PtdIns(3,5)*P*_2_, PtdIns(4,5)*P*_2_, and PtdIns(3,4,5)*P*_3_ [7].

PIs are minor components of the phospholipid bilayer of cellular membranes throughout Eukarya (only 1% of the total lipid pool) [12]; however, they are vital as second messengers regulating diverse cellular functions, including all major signal transduction pathways, cytoskeleton dynamics, and intracellular membrane traffic for endocytosis and autophagy [13,14]. Signaling by PIs involves an array of protein effectors that recognize and bind individual PIs through lipid recognition domains, i.e., PH (pleckstrin homology), ENTH (epsin amino-terminal homology), FYVE (named after the four cysteine rich proteins in which it has been found: Fab 1, YotB, Vac1p, and EEA1), PX (the phox homology domain), C2, and others [15,16]. Deregulations of PI metabolism or PI-based signal transduction networks are responsible for a number of human diseases. Moreover, a number of pathogens hijack the PI-based regulatory systems of host cells. As a result, PI-converting enzymes and PI-interacting proteins with lipid-binding domains are being noticed as potential therapeutic targets [2,17].

The members of the SNX-BAR subfamily of sorting nexins, aside from the PX domain, contain an additional BAR domain, a dimerization structural motif that induces membrane deformation, thereby transitioning flat membranes to tubular membrane surfaces [18]. Current models propose that both PX and BAR domains have to be engaged with the membrane to ensure specificity and the efficient binding of this class of PI-binding proteins. Mammalian cells possess twelve SNX-BAR subfamily members [19]. SNX1, SNX2, SNX5, and SNX6 are critical for endosome-to-TGN retrieval and endosome-to-plasma membrane recycling [20,21], while other SNX-BAR proteins have retromer-dependent or -independent roles in protein sorting or autophagic processes [22,23].

An intricate competition exists between pathogens and their hosts [24,25,26] in controlling cellular trafficking regulated by SNXs. Several intracellular pathogens hijack SNXs as a part of the mechanisms used to support their intracellular survival without themselves expressing SNXs. Such examples are the bacteria *Salmonella typhimurium* and *Chlamydia trachomatis*, which hijack the roles of SNX18/SNX3 and SNX5/SNX6, respectively [27,28,29,30]. *Legionella pneumophila* also inhibits retrograde trafficking, regulated by retromer SNXs, to promote its intracellular replication [31]. Similarly to pathogenic bacteria, eukaryotic pathogens, such as the oomycete *Phytophthora infestans* and the apicomplexan *Plasmodium falciparum*, appear to utilize PtdIns3*P* at the host plasma membrane and the parasites’ endoplasmic reticulum (ER) lumen to modulate endocytic and exocytic trafficking pathways, respectively, for both secretion and pathogenesis [32,33].

PI-signaling and PI-interacting proteins are very poorly investigated in parasitic protozoans [32,34]. The latter possess restricted repertoires of PI-binding proteins associated with reduced or modified pathways that are often related to atypical endomembrane compartments. Adaptations to narrow biological niches have resulted in either innovation in existing organelles and/or the likely repurposing of endosomal organelles [35]. For these reasons, parasitic protists have often served as model organisms to investigate the evolutionary origins and minimal essential components of vesicle trafficking/membrane transport in extant eukaryotes as well as in hypothetical models for the last eukaryotic common ancestor (LECA) (recently reviewed in [32,35,36]). In the *Leishmania* species (spp.), the flagellated protozoans of the Kinetoplastea class from the early eukaryotes’ Euglenozoa phylum [37], a phylogenetic analysis has revealed the genes encoding four core eukaryotic PI kinases, type III PIK4A and PIK4B, and at least one homolog each from the PI3K (possibly PIK3C3 as the ancestor) and PIP5K families, supporting the existence of second messenger signaling based on PIs in these protozoan parasites [32,38]. Moreover, in the *Trypanosoma* spp., the PtdIns(4,5)*P*_2_ is enriched at the cytosolic face of the flagellar pocket, the cell region where endocytosis and exocytosis occurs [39]. Since most of the Trypanosomatidae members are obligate intracellular pathogens (i.e., *Leishmania* and *Trypanosoma* spp.), the identification and study of PI-interacting proteins involved either in the signal transduction pathways mediated by PIs or in PI metabolism could highlight potential anti-parasitic drug targets [17,40], given the extremely important role of PI-interacting proteins in cellular processes that are fundamental to Eukaryota life [41].

The epicenter of this study is the molecular and functional characterization of the *Leishmania* LDBPK_352470.1 gene product, registered in UniProt KB under the identifier A0A504X805. Although it is designated as a putative PI-binding protein in UniProt and as autophagy-related protein 24 in TriTrypDB, herein, we provide grounded evidence that it is a sorting nexin retromer component, the first ever characterized in *Leishmania* spp.

## 2. Results

### 2.1. The LdBPK_352470.1 Gene Product Is Highly Conserved in Leishmania spp. and Has Orthologs in Trypanosoma spp.

A search in databases revealed that the product of the LDBPK_352470 gene is registered in UniProt under the identifier E9BSB7_LEIDB. This registration was merged recently and now is designated as A0A504X805 and is characterized as a putative phosphoinositide-binding protein. In the TriTryp database, this protein is registered under the accession LdBPK_352470.1 and has been annotated as autophagy-related protein 24. E9BSB7_LEIDB is the ortholog of the LmjF.35.2420 gene product attributed to a secreted protein initially identified in a previous proteomic analysis of the *L. donovani* secretome by Silverman et al. 2008 [42]. This protein was initially classified as a phosphoinositide-binding protein (PIBP) [42]. The high abundance of this PIBP in the secreted cellular fraction is suggestive of a protein having a role in the parasite–host interaction. Given the limited knowledge on proteins involved in the PI regulatory system of *Leishmania* parasites, we were prompted to investigate the structural and functional traits of this *L. donovani* putative PIBP. 

Initially, the sequence of the *Ld*BPK_352470 gene was used as the input in a search within the TryTripDB. As shown in Table 1, 21 *Leishmania*, 2 *Leptomonas*, and 26 *Trypanosoma* predicted protein orthologs were retrieved. Βlastp pairwise sequence alignment revealed significantly high sequence conservation among A0A504X805 (E9BSB7_LEIDB) and the 21 orthologs from 14 *Leishmania* spp. and the 2 *Leptomonas* orthologs with sequence identities of 89.5–100%, query covers of 99–100%, and E-value = 0.0. Similarly, significant sequence conservation was observed between A0A504X805 (E9BSB7_LEIDB) and 23 out of the 26 *Trypanosoma* orthologs (sequence identity 31.00–34.67%, query cover 96–98%, E-value < 4 × 10^−64^. All *Leishmania* orthologs consist of 417 or 418 aa residues, while the length of *Trypanosoma* orthologs span from 415 to 429 aa (Table 1). A comprehensive phylogenetic tree showing close and distant evolutionary relationships between the A0A504X805 and its orthologs in *Leishmania* and *Trypanosoma* spp. is depicted in Figure 1.

### 2.2. Structure Prediction of A0A504X805 Reveals Domains That Classify It to the BAR Subfamily of Sorting Nexins

To investigate the structural components of the A0A504X805 (E9BSB7_LEIDB) protein, its amino acid sequence was used as an input within UniProt, Pfam, InterPro, and Prosite databases. Predictions from all databases revealed the same two structural domains. A Phox homology (PX) domain, comprising three β-sheets followed by three α-helices, was predicted between aa 8 and 139, in the N-terminal half of this protein, and a BAR/Vps5 domain, consisting of three banana-shaped helices, was predicted between aa 211 and 406 in its C-terminal half (Figure 2A).

Secondary structure prediction using the PsiPred algorithm verified the existence of three β-sheets followed by three α-helices, indicative of the PX domain architecture. The application of the PsiPred algorithm also confirmed the existence of the BAR domain’s three helices (Appendix A). Tertiary structure prediction, with data from Alpha Fold, using the PyMol tool revealed the characteristic structures of both PX and BAR/Vps5 domains, as shown in Figure 2B. A coiled coil region (64–72 amino acid long) separating the two domains (Figure 2B and Appendix A) was also predicted both in UniProt and by the PsiPred algorithm. This predicted PX-BAR/Vps5 architecture classifies the A0A504X805 to the sorting nexin–BAR subfamily of SNXs [2]. The Vps5 protein, containing both PX and BAR domains, is the yeast counterpart of human SNX1 and is part of the retromer complex. However, the term Vps5 is often used in many databases as an alternative name for the BAR domain [43]. Based on the results of the above analyses, we named this phosphoinositide-binding protein (PIBP), encoded by the *Ld*BPK_352470 gene, *Ld*SNXi [*Leishmania* (*L.*) *donovani* Sorting Nexin i (number 1 in latin)]. This is the first sorting nexin to be identified in *Leishmania* protozoan parasites.

### 2.3. Cloning of the LdBPK_352470 Gene from L. donovani Genomic DNA and Generation of Specific Molecular Reagents

To biochemically and functionally characterize *Ld*SNXi, several specific molecular reagents were generated. Initially, the *Ld*BPK_352470 gene was cloned into the pGEX4T-1 and pEGFP-N3 vectors through a PCR-based approach using genomic DNA from the *L. donovani* LG13 strain (MHOM/ET/0000/HUSSEN). Thereby, the pGΕΧ-4Τ1-*ldsnx_i_* and pEGFP-*ldsnx_i_* plasmids were generated (Section 4, Appendix A). The sequencing of the inserted DNA sequences showed 100% identity to the LdBPK_352470 gene sequence registered in the TriTrypDB (Appendix A, *Ld*SNXi sequence).

To confirm the predicted binding of *Ld*SNXi to PIs, as expected for a PX domain-containing protein, and determine its PI substrate specificity, the pGEX-*ldsnx_i_* recombinant plasmid was used to express the full-length *Ld*SNXi as a Glutathione-S- Transferase (GST) hybrid protein (GST-*Ld*SNXi) in bacteria (Appendix A). Furthermore, the pEGFP-*ldsnx_i_* plasmid was used to transfect mammalian HeLa epithelial cells to further analyze the subcellular localization of *Ld*SNXi-EGFP by confocal microscopy.

Two more plasmids were also constructed, the pGΕΧ-4Τ1*-ldsnx_i_-c-term* and the pMAL-C2-*ldsnx_i_-c-term*, encoding the C-terminal domain of *Ld*SNXi (a.a. 264–417) giving rise to recombinant fusion proteins with GST or Maltose-Binding Protein (MBP), respectively, at their N-terminus (schematic representation in Appendix A). The MBP-*Ld*SNXi-C-term protein (Appendix A) was used for rabbit immunization to generate a-*Ld*SNXi-specific sera. The GST-*Ld*SNXi-C-term (Appendix A) was used to verify the specificity of a-*Ld*SNXi-specific Abs (Appendix A) and purify an antibody pool enriched in a-*Ld*SNXi specific Abs. The sequence of the *ldsnx_i_* gene in the constructed plasmids was verified by sequencing. The expression of the correctly sized recombinant proteins was checked by SDS-PAGE and Western blot (Appendix A).

### 2.4. LdSNXi Is Expressed in L. donovani as a Soluble Cytosolic and as Secreted Protein

To confirm *Ld*SNXi expression in *L. donovani* and assess its abundance at different developmental stages, promastigote cultures of the *L. donovani* LG13 strain were used. These promastigotes, cultured at 25 °C and pH 7, were analyzed at the logarithmic (enriched in dividing cells) and stationary (enriched in metacyclic promastigotes) phases of growth. Total lysates from parasite cultures were analyzed by Western blot using the purified a-*Ld*SNXi specific polyclonal antibody (pAb) (Section 4 and Appendix A). In all cases (i.e., logarithmic- and stationary-phase promastigotes growing at 25 °C and pH 7), the a-*Ld*SNXi pAb detected a protein zone migrating next to the 48 kDa molecular size marker (Figure 3A(a1) and Appendix A). Given that the calculated MW of *Ld*SNXi is 46.6 kDa, the protein species detected with the specific a-*Ld*SNXi Ab was considered to correspond to the nascent *Ld*SNXi. The analysis of this protein’s expression levels in promastigote cultures showed that it is expressed at approximately similar levels both in the logarithmic and stationary phases (Figure 3A). In a similar analysis in the dermatotropic *Leishmania* species *L. major*, the a-*Ld*SNXi Ab detected a protein species migrating with the same mobility as the protein detected in the viscerotropic species *L. donovani* (Figure 3A(a1)), a result confirming the conservation of this protein in the *Leishmania* genus.

To examine whether *Ld*SNXi associates preferentially with certain subcellular compartments, *L. donovani* promastigotes were subjected to subcellular fractionation by treatment with different detergents (Section 4) [44,45]. This procedure generated the following: subcellular fractions enriched in (a) soluble cytosolic proteins (Fractions F1 and F2); (b) proteins associated with intracellular organelles (Fractions F3 and F4), ERs (Fraction F4), and pelicular/surface membranes (Fraction F5s); and (c) an insoluble fraction (F5ins), as the final pellet, enriched in nuclear and cytoskeletal proteins. These fractions were further analyzed by Western blot using the specific a-*Ld*SNXi pAb. *Ld*SNXi was detected in the fractions enriched in soluble proteins (i.e., Fractions F1 and F2; Figure 3B). A higher-than-expected molecular size band was also detected in the insoluble cytoskeletal fraction.

The LDBPK_352470.1 gene product (i.e., herein *Ld*SNXi) was initially detected as a secreted *L. donovani* protein [42]. To confirm the *Ld*SNXi secretion by *L. donovani* promastigotes, stationary phase parasites were incubated in FBS-free RPMI at 25 °C for 6 h. Proteins in the extracellular medium were collected by acetone precipitation (Section 4) and further analyzed by Western blot side by side with the promastigote pellet proteins. *Ld*SNXi was detected by the specific a*-Ld*SNXi pAb in the extracellular and intracellular fractions (Figure 3C(a3)). Two other *Leishmania* proteins, the nuclear histone H2B (*Ld*H2B) and the cytoplasmic and membrane bound *Ld*TyrPIP_22 phosphatase that was recently characterized by our group [46], were used as controls. Both these proteins were detected only in the *Leishmania* promastigote pellet (Figure 3C(a3,c3)).

### 2.5. LdSNXi Binds Preferentially to PtdIns3P and PtdIns4P

The prediction of the *Ld*SNXi secondary structure showed that it contains a PX domain architecture, known to bind with preference to PtdIns3*P* but also to various di- and tri-phosphorylated phosphoinositides [7]. To biochemically confirm the predicted binding of *Ld*SNXi to PIs and its PI-binding preference, recombinant full-length GST-*Ld*SNXi expressed in bacteria transformed with the pGEX-*ldsnx_i_* plasmid was purified (Appendix A) and used in a PI-binding dot blot assay. Purified GST (Appendix A) was used as a negative control. Finally, conditions were established whereby the GST-*Ld*SNXi was expressed at about 40% as soluble protein (Appendix A). The proteins recovered in the soluble fraction after purification by affinity chromatography on glutathione agarose beads (Section 4, Appendix A) were used in the PI-binding assays. GST-*Ld*SNXi was detected to bind preferentially to PtdIns3*P* and PtdIns4*P* (Figure 4A) solidifying further the classification of *Ld*SNXi to the family of SNXs [7].

### 2.6. Endogenous LdSNXi Shows a Vesicular Localization Pattern in L. donovani Promastigotes and Axenic Amastigote-like Cells and Partially Co-Localizes with Microtubules

To deduce possible *Ld*SNXi functions, the subcellular localization of the endogenous protein was examined in *Leishmania* cells using immunofluorescence labeling and confocal microscopy imaging. For this, the a-*Ld*SNXi pAb was used in combination with an a-*Ld*Tubulin-specific antibody. Interestingly, in the dividing and non-dividing morphological forms of *L. donovani* promastigotes, *Ld*SNXi localized in vesicle-like structures in the cell body and the flagellum and at the periphery of the cells along subpellicular microtubules (Figure 5A). Ιn mitotic dividing promastigotes, *Ld*SNXi showed a vesicular pattern, also spreading in the intercellular bridge area, where strong microtubule staining was observed (Figure 5B). A similar localization to that of promastigotes was observed in axenic amastigote-like parasites (Figure 5D). *Ld*SNXi epitopes were observed in distinct vesicles in the cell body near the periphery of the cell (Figure 5D(i,ii)), while a lower intensity signal followed the microtubule network (Figure 5D(iii)).

Overall, in *Leishmania* cells, *Ld*SNXi presents vesicular localization partially co-localizing with microtubules. The extent of co-localization between *Ld*SNXi epitopes and microtubules in promastigotes (green and red FL respectively) was measured using the 3D “Coloc” module of Imaris v9.3.1 (Section 4). The thresholded Mander’s coefficients were taken into account. Depending on the signal pattern detected for tubulin, two different groups were highlighted (as described before by Ambit et al., 2011 [47]): (1) promastigotes with a long slender cell body where the subpellicular microtubules (ΜΤs) were labeled by the a-tubulin antibody; and (2) promastigotes with a wider cell body in which the posterior-end MTs were predominantly stained. We followed the naming proposed by Ambit et al. for “form a” parasites with the *Ld*Tubulin signal all around the periphery of the cell body and the flagellum (subpelicular MTs) and “ form a′ ” parasites where *Ld*Tubulin presented a localized, higher intensity signal at the posterior end of the cell body.

As shown in Figure 5C, in the bar graph depicting the thresholded Manders coefficients, the green FL pixels (thresholded Manders coefficient A, corresponding to the a-*Ld*SNXi staining) co-localized at 75.14 ± 5.09%, and the red FL pixels (thresholded Manders coefficient B, corresponding to the a-*Ld*Tubulin staining) co-localized at 48.94 ± 4.96% in “ form a′ ” promastigotes. We attribute this difference to the number of pixels corresponding to *Ld*Tubulin epitopes covering a smaller area of the total pixel count in “ form a′ ”. The high degree of co-localization, especially in “form a” promastigotes, corresponds to the observation that *Ld*SNXi epitopes appear as vesicular filamentous staining which follows MTs.

### 2.7. LdSNXi Is Predicted as the Evolutionary Structural Homologue of Human SNX1 and SNX2

Given that SNXs are conserved in eukaryotes, where their function has been greatly elucidated, a search was conducted for *Ld*SNXi homologs in higher eukaryotes to further infer their potential function(s). *Homo sapiens* is the organism where the most SNXs have been characterized thus far [7]. The consequent construction of a neighbor-joining phylogenetic tree comprising *Ld*SNXi and all human SNXs (32) (Figure 6) revealed that *Ld*SNXi is evolutionary closer to human SNX1 and SNX2. The pairwise sequence alignment of *Ld*SNXi and all human PX-BAR sorting nexins (eleven) using Blastp revealed a sequence conservation ranging from 23.76% to 36.00%, with a query cover from 13 to 96% (Table 2). Although SNX4 has the highest percentage identity (36%), this is restricted to a query coverage of 23%, corresponding, most probably, to a region spanning the PX domain (Table 2). Interestingly, for SNX1 and SNX2, although they present a lower percent identity to *Ld*SNXi (24.11% and 23.56%, respectively), the corresponding query coverage is as high as 96% (Table 2). Notably, SNX6 is not included in Table 2 because its homology is not statistically significant, according to the Blastp algorithm. According to these findings, *Ld*SNXi could be structurally closer to SNX1 and SNX2 than to SNX4, as reported in the TriTryp database.

This is further supported by the evolutionary distances between *Ld*SNXi and SNX1, SNX2, and SNX4, as depicted in the phylogenetic tree shown in Figure 6. *Ld*SNXi is located in the same clade as human SNX1 and SNX2 (together with SNX5, SNX6, and SNX32), while SNX4 belongs to a different and more distant clade.

This discrepancy in the structural homology between *Ld*SNXi and SNX4 (as reported in TriTrypDB) or SNX1/SNX2, as reported herein, prompted us to further elucidate the sequence homology of individual PX and BAR domains of *Ld*SNXi and human PX-BAR SNXs. Blastp alignment for isolated PX and BAR domains (Table 3) illustrated that although SNX4 shows higher percentage identity for its PX domain (38.30%) as compared to SNX1 and SNX2 (30.83% and 31.67%, respectively), the query coverage of PX_SNX4 is lower than that of PX_SNX1 or PX_SNX2 (Appendix A), corroborating that *Ld*SNXi is structurally closer to human SNX1/SNX2 than to SNX4.

This finding is further empowered by a Blastp pairwise sequence alignment of isolated BAR domains of *Ld*SNXi and human SNXs which returned only SNX2 as having significant homology to *Ld*SNXi (Table 3 and Appendix A). The 2D and 3D structure prediction analysis of *Ld*SNXi using PsiPred and PyMol, respectively, revealed the close structural homology of the PX and BAR domains of *Ld*SNXi and SNX1 (Appendix A). Unfortunately, the SNX4 structure has not been solved by crystallography yet to perform a similar comparison.

Given that SNX1 and SNX2 along with SNX5 and SNX6 are all components of the human retromer complex [48,49,50], and the fact that *Ld*SNXi is evolutionarily close to SNX1 or SNX2, one could postulate that *Ld*SNXi could have a similar function in a putative *Leishmania* retromer complex.

### 2.8. Multiple Localizations of the LdSNXi-GFP Heterologously Expressed in Mammalian Cells in Golgi, Microtubules, and the Vicinity of Mitotic Chromosomes

Since *Ld*SNXi is secreted in the extracellular medium of *Leishmania* promastigote culture, it is possible that it may play a role in the interaction of *Leishmania* parasites with host cells. In this case, it is worth exploring the possibility that it interacts with the human structural homologs highlighted in this study (i.e., SNX1 and SNX2 members of the retromer complex, SNX4, and SNX7) or their partners.

As a first step in this exploration, the localization pattern of heterologously expressed *Ld*SNXi as a recombinant EGFP hybrid (i.e., *Ld*SNXi-EGFP) was investigated in human HeLa cells transfected for 24 h with the pEGFP-*ldsnx_i_* plasmid. For this, immunofluorescence labeling was conducted for different cellular markers, and imaging was performed by confocal microscopy. A diffuse localization signal was observed for *Ld*SNXi-EGFP in cells without pre-extraction (Appendix A), confirming the finding in *Leishmania* cells where endogenous *Ld*SNXi was distributed mainly in the parasite’s subcellular fractions enriched in cytoplasmic-soluble proteins (Figure 3B). However, after a short pre-extraction step with 0.05% (*v*/*v*) Triton X-100 for the removal of soluble cytoplasmic proteins, vesicular, reticular, and filamentous *Ld*SNXi-EGFP localization patterns were clearly detected (Figure 7A). Moreover, *Ld*SNXi-EGFP partially co-localized with microtubules both at interphase and mitotic cells, as well as with the Golgi apparatus stained for Golgin97, a member of the Golgin protein family localizing at the Golgi membranes (Figure 7C). These localizations are compatible with the localizations of the retromer components [18,48].

Interestingly, in mitotic cells, *Ld*SNXi-EGFP was detected (a) in the vicinity of chromosomes (more intensely during prometaphase), (b) at the spindle microtubules faintly, and (c) at the spindle poles in metaphase cells (Appendix A). Moreover, in late anaphase and telophase, *Ld*SNXi-EGFP was also detected at the intercellular bridge (Figure 7D), a region of intense vesicular trafficking [51]. This confirmed the localization of endogenous *Ld*SNXi in the *Leishmania* dividing promastigotes (Figure 5B).

### 2.9. In Silico Analysis Reveals That LdSNXi Is Functionally Homologous to Human SNX1 and SNX2

To elucidate the functional correlation between *Ld*SNXi and human SNXs, a STRING database protein–protein interaction (PPI) network was constructed. Because no entry exists for the *Ld*SNXi (*Ld*BK_352470 gene product) in STRING db, its *Leishmania infantum* homolog (UniProt Accession: A4IBF2, further named *Lin*SNXi) identifier was used. According to STRING db PPI network (restricted to experimental evidence with high confidence interaction score, 0.900), *Lin*SNXi strongly interacts with Vps35, Vps26, and Vps29 (Metallophos_2), three out of five proteins known to form the retromer complex [52,53] (Figure 8A). Two other partners characterized by UniProt as “Derlins”, functional components of the degradation mechanism in the endoplasmic reticulum [54,55], were also retrieved as interacting with *Lin*SNXi (Figure 8A).

A query performed in the STRING database for the human sorting nexins SNX1, SNX2, SNX4, and SNX7 (medium confidence interaction score, 0.400), yielded four PPI networks (based only on experimental evidence). In all networks, the retromer complex members Vps35, Vps26A, Vps26B, and Vps29 were highlighted as interactors with all four human SNXs (Appendix A). It is intriguing that SNX1 and SNX2 were found to interact with high confidence (0.900) with all retromer components, which is comparable to *Lin*SNXi interactions with Vps35, Vps26, and Vps29-like (Metallophos_2) parasitic proteins. SNX4 and SNX7 displayed these interactions at lower confidence scores (0.400) (Figure 8B and Appendix A).

It is important to note that only the human SNX1 and SNX2, like the parasitic *Lin*SNXi, were found to interact with Derlin1, a membrane protein involved in the endoplasmic reticulum-associated degradation of misfolded proteins [54,55]. Taken together, the above findings suggest that *Ld*SNXi is functionally closer, in terms of molecular partner interactions, to SNX1/SNX2 than to SNX4.

## 3. Discussion

The present study reports the characterization of a novel member of SNXs in the ancient protists of the *Leishmania* genus. This new SNX was initially identified in a secretome analysis of *L. donovani* [42] as the *Ld*BPK_352470.1 gene product and shown herein to be highly conserved in *Leishmania* spp. Sequence analysis revealed a PX domain PI-binding structural module at the N-terminal half of this protein, confirming its initial classification as a phosphoinositide-binding protein [42]. PX domain-containing proteins are candidate PI effectors with roles in the PI regulatory network involved in fundamental processes for the life of eukaryotes [17,56,57], while all members of the SNX family contain at least one PX domain [2]. The *Ld*BPK_352470 gene encodes a protein of 417 a.a. residues (predicted MW 46.6 kDa) and is registered in UniProt under the identifier A0A504X805. Blastp pairwise sequence alignment, phylogenetic analysis, 2D and 3D structure predictions based on solved homologous structures, and STRING analysis indicated collectively that A0A504X805 is a member of the SNX-BAR subfamily of sorting nexins [18] and is predicted as the *Leishmania* homolog of human SNX2/SNX1 members of the retromer complex. The *Ld*BPK_352470.1 gene product is the first SNX to be identified in the *Leishmania* organism and was therefore named “*Ld*SNXi” for *Leishmania donovani*
Sorting NeXin i.

Sequence and structural homology analysis revealed that the closest human homolog of *Ld*SNXi is SNX2, in terms of PX- and BAR-domain conservation and PI-binding specificity. Thus far, it was presumed that *Ld*SNXi is the homolog of human SNX4. The findings presented herein show that although there is high homology between the *Ld*SNXi PX domain and the PX domains of SNX4 and SNX2 (38.3% and 31.67, respectively), statistically significant homology for the BAR domain of *Ld*SNXi was found only in SNX2. The biochemical data presented in this study support that *Ld*SNXi binds specifically to PtdIns3P and PtdIns4P. The SNX4 PX domain has been shown to bind specifically to PtdIns3P and PtdIns(3,4)P_2_/PtdIns(4,5)P_2_ [58]. SNX2 PX was shown to bind PtdIns(3,5)P_2_ in addition to PtdIns3P, while it also associates with lower affinity to PtdIns4P and PtdIns(3,4)P_2_ [59]. However, in a more recent study, the PX domains of SNX4 and SNX7 were shown to have very low affinity to PI3P, while SNX2 was shown to have a preference for binding to PtdIns(3,4)P_2_ [9]. In another earlier study though, VPS5, the yeast homolog of SNX1/SNX2, showed a preference for PtdIns3P binding [58]. To the authors’ knowledge, there is no study about the PI specificity of the VPS5 homolog in *T. brucei* [34] which would shed light into a different PI association of these SNXs in Trypanosomatidae. Τherefore, *Ld*SNXi, in terms of PI binding specificity, behaves partially as SNX4, as indicated by its higher homology to the SNX4 PX domain; however, it also binds to PtdIns4*P*, suggesting an overall functional homology to SNX2.

Intracellular *Ld*SNXi presented a vesicular/tubular localization pattern both in *L. donovani* (as endogenous) and in mammalian cells as heterologously expressed r*Ld*SNXi-GFP. Moreover, *Ld*SNXi vesicular staining co-localized with microtubules both in *Leishmania* and mammalian cells transfected with the pEGFP-*ldsnxi* plasmid. The mammalian retromer, which includes SNX1 or SNX2 and SNX5 or SNX6 [11,59,60,61], is coupled to the dynein microtubule motor complex via an interaction with the p150^Glued^ subunit of the dynein accessory complex dynactin [62,63]. This coupling is required for both the formation and fission of tubular carriers. Therefore, the *Ld*SNXi localization is compatible with the predicted retromeric localization and function. Moreover, the results from the structural homology and P–P interaction network analysis also predict that *Ld*SNXi is the structural and functional homolog of the human SNX2 protein. Taken together, all the above findings support the claim that *Ld*SNXi is a component of the *Leishmania* retromer complex.

The retromer complex is highly conserved in eukaryotes; homologs have been found in yeast, *C. elegans*, mouse, and human models [64,65,66,67]. In yeast, the retromer complex consists of five proteins: Vps35p, Vps26p, Vps29p, Vps17p, and Vps5p. The mammalian retromer consists of the Vps26, Vps29, and Vps35 mammalian homologs and the SNX1*/*SNX2 and SNX5/SNX6 homologs [60]. It is proposed to act in two subcomplexes: (1) A cargo recognition heterotrimeric complex consisting of Vps35, Vps29, and Vps26; and (2) SNX-BAR dimers, which consist of SNX1 or SNX2 and SNX5 or SNX6 and facilitate endosomal membrane curvature remodulation, resulting in the formation of tubules/vesicles that transport cargo molecules to the trans-Golgi network [59,68]. The retromer complex has a well-established role in endosomal protein sorting, being necessary for maintaining the dynamic localization of hundreds of membrane proteins traversing the endocytic system. Several components of the retromer complex have been identified in *Trypanosoma* (*T.*) *brucei*, which is closely related to *Leishmania* Kinetoplastida, including the homologs of the yeast VPS35, VPS26, VPS5, and VPS17 [34]. The *T. brucei* VPS5 is the *Trypanosoma Ld*SNXi homolog (Figure 1) as shown in the work by Petsana et al. 2023 [8]. The fact that human SNX1, SNX2, SNX4, and SNX7, which are hierarchically the closest human homologs of *Ld*SNXi, are all predicted to interact with Vps26, Vps29, and Vps35 in the PPI networks highlights their structural and functional homology. More important, though, is the prediction that only *Ld*SNXi, SNX1, and SNX2 interact with Derlin or Derlin-homologous proteins, indicating a higher functional homology of these three SNXs. Taken together, our results not only highlight that *Leishmania Ld*SNXi is the structural and functional homolog of human SNX2 but also denote a high degree of conservation of the retromeric function in the distinct eukaryote evolutionary branch of Kinetoplastida.

Besides the localizations observed for *Ld*SNXi, which are compatible with a putative role as a *Leishmania* retromer component, *Ld*SNXi and r*Ld*SNXi-EGFP were also detected in the intercellular bridge [51] in *Leishania* and mammalian cells, respectively, and on the mitotic apparatus and around chromosomes in mammalian cells (r*Ld*SNXi-EGFP). These findings suggest the possible role(s) of *Ld*SNXi in regulating the events of endosomal organization or cargo sorting during different steps of mitosis. Cell division involves considerable membrane remodeling and vesicular trafficking (recently reviewed by [51,69,70]). Endocytosis and endocytic proteins are required for efficient mitotic progression and completion. Several endocytic proteins have been shown to participate in mitosis in an endocytosis-independent manner. The SNX9 subfamily members—SNX9, SNX18, and SNX33—are reported to be required for the progression and completion of mitosis through both endocytosis-dependent and -independent processes [71]. Perturbing endocytosis by various means leads to cytokinetic defects, presumably due to a reduced amount of recycling endosomes [51]. Taken together, our data suggest that *Ld*SNXi, as an ancient eukaryotic precursor of the numerous human SNXs, may have multiple cellular functions, including roles in mitosis, which in animal cells are carried out by different SNXs. During evolution, several of these functions could have been transferred to different SNXs in higher eukaryotic organisms. Interestingly, as shown in our recently published work [8], *Ld*SNXi is the only SNX-BAR protein identified in *Leishmania* spp. genomes.

Another interesting finding of this work was the detection of *Ld*SNXi in the extracellular medium of stationary-phase *L. donovani* promastigotes, a parasite pool enriched in the infectious metacyclic form of *Leishmania*. Although this was also the initial finding of the proteomic study by Silverman et al. [42], where *Ld*SNXi was highlighted as one of the most abundant secreted proteins, more recent studies on the secretome of seven different *Leishmania* spp. do not mention this protein as secreted [72], while an exosome content analysis in *L. infantum* detected low levels of the *Lin*SNXi in logarithmic phase parasites’ exosomes [73]. These conflicting results may be due to the different approaches followed for extracellular medium manipulation, as well as to different cut-off limits applied as to what material was considered secreted in each case. Our results are in accordance with two out of the three studies on the secretome of *Leishmania* spp. Therefore, it is highly possible that *Ld*SNXi is secreted by the parasites. In that case, *Ld*SNXi may play a role in the interaction of *Leishmania* with host cells, a hypothesis yet to be confirmed by future experimental evidence.

## 4. Materials and Methods

Reagents and antibodies: All chemicals, unless otherwise stated, were of analytical grade and purchased from Sigma-Aldrich (Darmstadt, Germany) or Applichem GmbH, (Darmstadt, Germany). Restriction enzymes were purchased from Roche (New England Biolabs, Ipswich, MA, USA) and/or KAPA Biosystems (Sigma-Aldrich, St. Louis, MO, USA). Taq DNA polymerase (R001A) and T4 ligase (2011B) were from TaKaRa (Kusatsu, Shiga, Japan). All primers used in the PCR reactions (synthesized by VBC Biotech, Vienna, Austria) are listed in Appendix A. DNase I (2270A) was from TaKaRa, and RNAse (10109134001) was from Sigma Aldrich (Merck, St. Louis, MO, USA). DNA ladder of 1 Kb (N3232L) was from New England Biolabs (NEΒ, Ipswich, MA, USA) or from Nippon Genetics (Düren, Germany; MWD1P). Protein molecular mass standards (17-0446-01) were purchased from Amersham Biosciences (Amersham, UK) and Nippon Genetics, Düren, Germany (Broad range:10–180 kDa, MWP03). Fetal bovine serum was from Thermo Fisher Scientific, Waltham, MA, USA (FBS, 10270106) or Biosera, Cholet, France (FB-1001/500). Bacto-tryptone (211705), Bacto Yeast extract (212750) and Bacto-agar (14050) were from BD Biosciences (Franklin Lakes, NJ, USA). The a-tubulin (T5168) mouse monoclonal (mAb) was from Sigma (St. Louis, MO, USA), the a-EF1a Ab clone CBP-KK1 (05-235) was from Merck (St. Louis, MO, USA), the a-Golgin97 mouse IgG (monoclonal CDF4; A-21270) was from Molecular Probes (Eugene, OR, USA) the polyclonal a-EEA1 rabbit IgG (ab137403) was from Abcam (Cambridge, UK) and the a-GST (A-5800) was from Invitrogen (Waltham, MA, USA). Fluorochrome-conjugated secondary Abs [CF546 and CF488 (20010 and 20012)], goat anti-rabbit HRP (20402) and goat anti-mouse HRP (20401) were from Biotium (San Francisco Bay Area, CA, USA). Hoechst 33342 (H3570) was purchased from Molecular Probes (Eugene, OR, USA).

### 4.1. Cell Culture

The human epithelial cell line HeLa was cultured in low-glucose DMEM (Biosera PM-D1105) containing 10% (*v*/*v*) hiFBS [heat inactivated (at 56 °C for 30 min) fetal bovine serum], 1 U/mL penicillin, and 0.1 mg/mL streptomycin. Mammalian cells were counted with a Neubauer hemocytometer (Kyrios Soter Scientific, Miami, FL 33186 USA).

*L. donovani* (strain LG13, MHOM/ET/0000/HUSSEN) promastigotes were cultured in RPMI 1640 containing 10% (*v*/*v*) hiFBS, 1 U/mL penicillin, 0.1 mg/mL streptomycin (Gibco, Thermo Fisher Scientific, Waltham, MA, USA), and 10 mM Hepes (Gibco, Thermo Fisher Scientific, Waltham, MA, USA) at 25 °C, as previously described in Papadaki et al. [46]). *Leishmania* cells were counted with a Malassez hemocytometer. *L. donovani* axenic amastigotes were obtained according to a modified published protocol [46].

### 4.2. Cloning and DNA Constructs

The sequence of the LdBPK_352470 (ΤriTrypDB) gene encoding the *Ld*SNXi unique C-terminal sequence [796–1254 base pairs (bp), 264–417 amino acids (aa)] was amplified by polymerase chain reaction (PCR) from genomic *L. donovani* DNA (strain LG13) using primers 4 and 5 (Appendix A) and inserted into the *EcoRI* site of the pMAL-c2 and pGEX-4T1 vectors, in frame with the ORF encoding the maltose-binding protein (MBP; pMAL-c2, New England Biolabs Inc.) or glutathione transferase (GST; pGEX-4T1, Pharmacia) to produce the pMAL-c2-*ldsnx_i_-c-term* and pGΕΧ-4Τ1*-ldsnx_i_-c-term* plasmids, respectively. The gene encoding the full-length *Ld*SNXi [1–1254 bp, 1–417 aa, LdBPK_352470] was amplified by PCR from genomic *L. donovani* DNA (strain LG13) using primers 3 and 4 (Appendix A) and inserted into the *EcoRI* site of the pGEX-4T1 vector, in frame with the GST-encoding open reading frame (ORF), to produce the pGEX-4T1-*ldsnx_i_* plasmid. Moreover, the gene encoding the full-length *Ld*SNXi (LdBPK_352470) was amplified by PCR, as above, using primers 1 and 2 (Appendix A) and inserted into the *BglII*/*BamHI* site of the pEGFP-N3 vector, in frame with the ORF encoding the enhanced green fluorescent protein (EGFP), to produce the pEGFP-N3-*ldSNXi* plasmid. All plasmid constructs were propagated in the *Escherichia coli* (*E. coli*) XL1-Blue strain for plasmid DNA preparations. Two positive clones were selected and sequenced in each case (VBC-Biotech Services GmbH, Vienna, Austria) and/or in the Genomic Analysis and Bioinformatics Applications Unit, Hellenic Pasteur Institute). For recombinant protein production with the pMAL-c2 vector, *E. coli* (DH5 strain) clones harboring the appropriate plasmid were used, while for expression from pGΕΧ-4Τ1 based plasmids, *E. coli* BL21 (DE3 strain) clones harboring the appropriate plasmids were used.

### 4.3. Overexpression and Purification of Recombinant LdSNXi Forms

For GST-*Ld*SNXi-C-term (264–417 aa of *Ld*SNXi) expression, 100–1000 mL culture from the *E. coli* clone carrying the pGΕΧ-4Τ1*-ldsnx_i_-c-term* was incubated overnight (~16 h, OD_600_ ~ 1.6) at 25 °C and then induced with 0.1 mM isopropyl *β*-D-thiogalactoside (IPTG) at 25 °C for 3 h.

For GST-*Ld*SNXi full-length protein expression, a 10–100 mL starter culture from an *E. coli* clone carrying the pGEX-4T1-*ldsnx_i_* plasmid was incubated at 25 °C overnight, diluted to OD_600_ ~ 0.4 the following morning, incubated at 25 °C for 2–3 h until OD_600_ = 0.5–0.6, and then induced with 0.6 mM IPTG for 3 h. *E. coli* cells were harvested by centrifugation (4000× *g*, 15 min). To produce the recombinant MBP fusion protein for rabbit immunization, a 2 L culture (OD_600_ = 0.5–0.6) of an *E. coli* clone carrying the pMAL-c2-*ldsnx_i_-c-term* plasmid, generated by diluting a 200 mL overnight starter culture grown at 37 °C, was induced with 0.5 mM IPTG at 37 °C for 4 h.

In all cases, bacteria pellets were subsequently resuspended in lysis buffer (20 mM Tris-HCl, 200 mM NaCl, 1 mM EDTA, pH 7.5 for the MBP fusion protein and PBS, pH 7.4 for the GST fusion proteins) containing proteolytic inhibitors (1:100 from stock; Sigma P 2714). They were lysed by a freeze–thaw process repeated three times and followed by 6–8 sonication cycles (30–60 s, 100 W), with a 30 s intermediate step of pause and incubation in ice after each sonication step.

The recombinant MBP*-Ld*SNXi-C-term protein was purified by affinity chromatography using amylose resin (NEB BioLabs, E8021L) according to the manufacturer’s instructions. The GST-*Ld*SNXi and GST proteins were purified by glutathione–agarose affinity chromatography (New England BioLabs, Ipswich, MA, USA). The soluble fraction of the bacterial lysate was incubated with glutathione–agarose gel for 30 min. The glutathione–agarose gel with bound proteins was washed twice with PBS 1% *v*/*v* Triton X-100, and bound proteins were eluted with 10 mM glutathione dissolved in 50 mM Tris-HCl (pH 9).

### 4.4. Generation and Purification of Polyclonal Antibodies

Purified MBP-*Ld*SNXi-C-term protein fractions (Appendix A) were injected into New Zealand white rabbits to raise polyclonal antisera according to published protocols [74,75]. All experimental procedures for rabbit immunizations were approved by the Institutional Animal Bioethics Committee following the EU Directive 2010/63 and the National Law 2013/56. Specific a-*Ld*SNXi polyclonal Abs were purified by incubating sera from immunized rabbits with PVDF membranes with GST-*Ld*SNXi-C-term and further eluted by low pH (50 mM glycine, pH 2.8), as previously described [76]. The eluted antibody solutions were dialyzed over PBS (2 dialysis steps, 4 °C), and their protein concentration was determined by measuring OD_280_, further aliquoted in 100 μL, then snap frozen in liquid nitrogen and stored at −80 °C.

### 4.5. Phosphoinositide Binding Assay

For this protein–lipid overlay assay, PIP Membrane Strips (Invitrogen™ P23751) were used to determine the GST-*Ld*SNXi full-length lipid specificity. Each strip, after an initial blocking step (1 h, RT) with 5% (*w*/*v*) bovine serum albumin (BSA) in TBST [Tris-buffered saline (TBS; 50 mM Tris-Cl, pH 7.5, 150 mM NaCl) with 0.05% (*v*/*v*) Tween-20], was overlaid with equimolar solutions of either purified recombinant GST-*Ld*SNXi [50 ng/mL in 5% (*w*/*v*) BSA, TBST] or 17 ng/mL of purified GST (negative control) followed by a 3 h incubation at RT. Subsequently, the strips were washed three times (RT, 10 min) with TBST followed by a second protein-binding step (4 °C, overnight) with GST-*Ld*SNXi [50 ng/mL, in 5% (*w*/*v*) BSA, TBST] or GST (17 ng/mL) and 3 washings with TBST. The bound protein was further detected by the a-GST (0.5 μg/mL) rabbit pAb (A5800; Invitrogen, Waltham, MA, USA) following the Western blot and development by ECL protocols described below.

### 4.6. Detergent-Based Protein Fractionation

Digitonin permeabilization of stationary phase *L. donovani* promastigotes was based on protocols previously described [44,45,46] with slight modifications. Briefly, *L. donovani* (*wt* or transgenic) promastigotes (~2 × 10^9^) were harvested by centrifugation (1000× *g*, 7 min, 4 °C), washed twice in resuspension buffer (145 mM NaCl, 11 mM KCl, 75 mM Tris-HCl, pH 7.4) and, resuspended in 0.5 mL of the same buffer supplemented with protease inhibitors. Membrane permeabilization and protein fractionation were achieved by adding 0.5 mL digitonin solution prewarmed at 37 °C to progressively increased detergent concentrations (stepwise, 4 steps) to achieve digitonin concentrations of 10 μM, 100 μM, 0.5 mM, or 5 mM. In each step, a soluble fraction and a corresponding insoluble pellet were recovered. The final pellet, recovered after treatment with 5 mM digitonin (i.e., F5; enriched in plasma membrane, nuclei, and cytoskeletal proteins), was further solubilized with 0.5 mL 1% (*v*/*v*) TritonX-100 (1 h, 4 °C) and the soluble fraction (F5s) was recovered from the insoluble fraction (F5ins) by centrifugation (20,000× *g*, 20 min, 4 °C). The soluble fractions (F1–F4 and F5s) were subjected to acetone precipitation by the addition of acetone pre-chilled at −20 °C (volume equal to four sample volumes) followed by 1 h incubation at −20 °C. The protein pellets recovered and the F5ins fractions were solubilized in the Laemmli buffer, boiled (5 min, 95 °C), and further analyzed by SDS-PAGE and Western blot.

### 4.7. Protein Analysis by SDS-PAGE and Western Blot

Promastigotes’ cell lysates for protein analysis were prepared as described by Papadaki et al. [46]. Protein samples were analyzed by SDS-PAGE [12% (*w*/*v*) or 10% (*w*/*v*)] gel) and transferred to Hybond-C nitrocellulose (Amersham, Amersham, UK) membrane using a wet blotting apparatus (BioRad, Hercules, CA, USA). After protein transfer, Hybond-C membranes were stained with Ponceau S solution [0.5% (*w*/*v*) Ponceau S dissolved in 1% (*v*/*v*) acetic acid]. Nonspecific sites for Ab binding on the nitrocellulose membrane were blocked by incubation (1 h, RT) with blocking buffer [TBST, 5% (*w*/*v*) BSA]. Incubation with the primary Abs was performed overnight (~16 h) at 4 °C. The affinity-purified a-*Ld*SNXi rabbit pAb, diluted in TBST, was used at ~0.5 μg/mL. Other Abs were used as indicated in figure legends. After three washes in TBST, the blots were incubated (1.5 h, RT) with HRP-labeled a-mouse or a-rabbit Abs used at 1:8000 dilution. Following three washes in TBST and one final wash in TBS, Ab reactivity was revealed either by the ECL plus system (Amersham, Amersham, UK) or by the chromogenic DAB method. In the former case, membranes were exposed to Kodak (Rochester, NY, USA) photographic films further developed with Kodak reagents. Digital images were acquired by scanning the films with a conventional scanner. Re-probing with the a-EF1A (1:10,000) or a-Histone, when required, was performed after stripping the membrane by incubating twice (10 min, RT) with a low-pH buffer [1.5% (*w*/*v*) glycine, 0.1% (*w*/*v*) SDS, 1% (*v*/*v*) Tween 20, pH 2.2], followed by washing and re-blocking with the blocking buffer.

### 4.8. Immunofluorescence

*L. donovani* promastigotes were fixed with 2% (*w*/*v*) paraformaldehyde (PFA) (20 min, RT) in PBS, allowed to adhere to poly-L-lysine coated coverslips, and treated with 50 mM NH_4_Cl in PBS (10 min, RT), followed by one wash with PBS. Adherent HeLa cells were washed once with PBS containing Mg^2+^ and Ca^2+^ and then fixed (20 min, RT) with PFA [4% (*w*/*v*) in PBS]. The non-reacted PFA was neutralized with 50 mM NH_4_Cl in PBS (10 min, RT) followed by one wash with PBS. Fixed cells were incubated (1 h, RT) with the primary Abs in blocking buffer [PBS, 1% (*w*/*v*) BSA] and, after extensive washing, with the appropriate secondary Abs conjugated to CF^®^ 633, CF^®^ 546, and CF^®^ 488 at a final concentration of 2 μg/mL in blocking buffer (1 h, RT). Secondary Abs were removed with extensive washing and the DNA was stained (10 min, RT) with Hoechst 33342 (10 μg/mL). Coverslips were mounted on microscope slides with Mowiol 4–88 [10% (*w*/*v*) Mowiol in 100 mM Tris-HCl (pH 8.5), 25% (*v*/*v*) glycerol], sealed with nail polish, and stored at 4 °C.

### 4.9. Transfection of Mammalian Cells

Adherent HeLa cells plated on coverslips placed in 24-well plates were transfected at a 70–90% confluency using the Polyplus jet optimus reagent kit (101000051; Polyplus, Strasbourg, France) and 500 ng of plasmid DNA, according to the manufacturer’s instructions. After 24 h, the cells were washed once with PBS containing Mg^2+^ and Ca^2+^ and fixed (20 min, RT) with PFA [4% (*w*/*v*) in PBS]. The non-reacted PFA was neutralized with 50 mM NH_4_Cl in PBS (10 min, RT) and the cells were stained with primary and secondary Abs or phalloidin-CF633^®^. Coverslips were mounted as described in the previous paragraph. In some cases, a pre-extraction step was used, by treating (30 s, RT) cells with 0.05% (*v*/*v*) Triton X-100 in PBS followed by PFA fixation.

Microtubules were depolymerized by a step of Nocodazole treatment. After transfection of HeLa cells, the medium was changed to culture medium containing 10 μΜ Nocodazole (FUJI FILM Wako, Osaka, Japan), pre-warmed at 37 °C. The cells were then incubated at 37 °C, 5% (*v*/*v*) CO_2_ for 4 h. At the end of this period, the pre-extraction step was applied, followed by fixation and immunofluorescence staining.

### 4.10. Imaging by Confocal Microscopy

The microscopic analysis of *Leishmania* or transfected HeLa cells was performed with the Leica SP8 confocal microscope (Leica, Wetzlar, Germany) using the 63× apochromat lens. Image acquisition included the collection of z stacks of 0.3 for parasites or of 0.5 μm step size for transfected HeLa cells.

### 4.11. Co-localization Analysis

The extent of co-localization between *Ld*SNXi epitopes and tubulin of the *Leishmania* microtubules (green and red FL respectively) was measured using the 3D “Coloc” module in Imaris v9.3.1, which utilizes the algorithms introduced by Costes et al. for automatic threshold selection of the image channels [77]. The green channel was used as a masking area for the entire analysis to exclude background pixels from the co-localization analysis datasets (threshold value of 7). The mask channel was used in conjunction with the automatic threshold function. This way, Imaris Coloc generates a new channel (the co-localization channel), which only contains voxels representing the co-localization between the red and green channels. The thresholded Mander’s coefficients were taken into account.

### 4.12. Secretion Assay

*L. donovani* promastigotes in logarithmic and stationary-phase cultures (50 mL) were enumerated using a Malassez hemocytometer (Fisher Scientific, Göteborg, Sweden) and harvested by centrifugation (1000× *g*, 10 min, RT). *Leishmania* cell pellets were resuspended in the same volume of RPMI, and the cells were collected by centrifugation (1000× *g*, 10 min, RT). Subsequently, the promastigotes’ pellet was resuspended in RPMI/10 mM Hepes (1/5th of the starter’s culture volume) without FBS and promastigotes were incubated at 25 °C and pH 7 for 6 h. At the end of the incubation period, the cells were separated from the culture supernatant by centrifugation, washed with PBS, and stored (−20 °C) until use. The culture supernatant was centrifuged (21,000× *g*, 20 min, 4 °C), filtered through a 0.2 μm filter to remove cell debris, and concentrated by acetone precipitation. Appropriate volume of Laemmli buffer 6× (20 μL) was added for the downstream process and samples were incubated for 30 min at 37 °C. Τhe protein content was estimated with the Macherey Nagel Protein Quantification Assay 740967.50 (Macherey Nagel, Düren, Germany).

### 4.13. Sequence Data Retrieval, Sequence Alignment, and Phylogenetic Tree Construction

The sequence of the *Ld*BPK_352470.1 gene product (*Ld*SNXi, A0A504X805 UniProtKB) as well as the sequences of its *Trypanosomatidae* orthologs were retrieved from the TriTryp database (TriTrypDB [78]). Pairwise sequence alignments of *Ld*SNXi with its orthologs were performed with the Blastp algorithm (v 2.13.0), and corresponding similarity scores were obtained. Multiple sequence alignment of the full-length amino acid sequences was performed with Clustal W (v 1.83) [79,80,81] with default settings (despite of branch length mode option which was disabled) and submitted to iTOL (v 6) [82,83] to generate a neighbor-joining phylogenetic tree. Amino acid sequences of the reviewed human PX-BAR sorting nexins were also manually obtained from [UniProt (2022_05)] [84,85].

### 4.14. Structure and Functional Predictionss

The 2D structure was constructed with PsiPred algorithm [86], using the *Ld*BPK_352470.1 gene product amino acid sequence, retrieved from TriTrypDB [78], as the input. The 3D structures of proteins which have not been solved were generated utilizing PyMol (The PyMOL Molecular Graphics System, Version 1.3, Schrödinger, LLC, New York, NY, USA), incorporating structure data from AlphaFold [87,88] (2021_07). The 3D-solved structures of the human SNX1 PX and BAR domains were constructed using PyMOL according to Protein Data Bank (PDB) data (registrations 2I4K and 4FZS) [89]. The Protein–Protein Interaction networks retrieved from the STRING database [90,91] were based only on experimental data.

## 5. Conclusions

This study provides valuable information on the evolutionary conservation of the SNX-BAR subfamily οf sorting nexins and forms the basis for future studies to further clarify their function in other eukaryotic pathogens. The additional functional analysis of the intracellular *Ld*SNXi in *Leishmania* cells using knock out or overexpressing genetically modified *Leishmania* strains will add insight into the functional evolution of eukaryotic SNXs, in particular to those forming the retromer complex, which has a well-established role in maintaining the dynamic localization of hundreds of membrane proteins traversing the endocytic system. The experimental validation of the 2D and 3D structural predictions of *Ld*SNXi PX and BAR domain structures by X-ray crystallography or nuclear magnetic resonance (NMR) spectroscopy will further elucidate the structural evolution of SNXs in eukaryotes. With respect to the *Ld*SNXi secreted pool, it remains to be shown whether it plays a role in the hijacking functions of host cells’ SNXs. Further studies using GST pull-down assays to experimentally identify *Ld*SNXi molecular partners in *Leishmania* and mammalian cells are in progress and will shed light on the functions of this novel SNX.

## Figures and Tables

**Figure 1 ijms-25-04095-f001:**
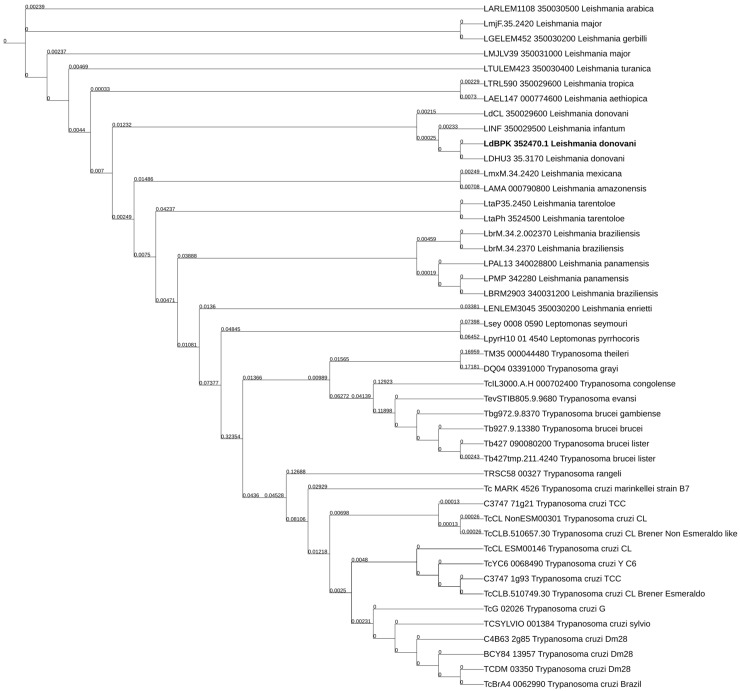
Evolutionary relationships of the *Ld*BPK 352470.1 gene orthologs in Trypanosomatids. Comprehensive neighbor-joining phylogenetic tree, generated using iTOL, based on multiple sequence alignment (ClustalW) of the *Ld*BPK 352470.1 gene product and its orthologs in *Leishmania* and *Trypanosoma* spp.; iTOL link: https://itol.embl.de/tree/1939213722931771674054468, accessed on 10 June 2023. Default scaling factors were used except for the vertical and horizontal scaling, which were adjusted to 1.2. Branch lengths corresponding to evolutionary distances are displayed on each corresponding branch of the phylogram. Statistically significant % identities of the sequences are shown in Table 1. *Ld*SNXi is depicted in bold.

**Figure 2 ijms-25-04095-f002:**
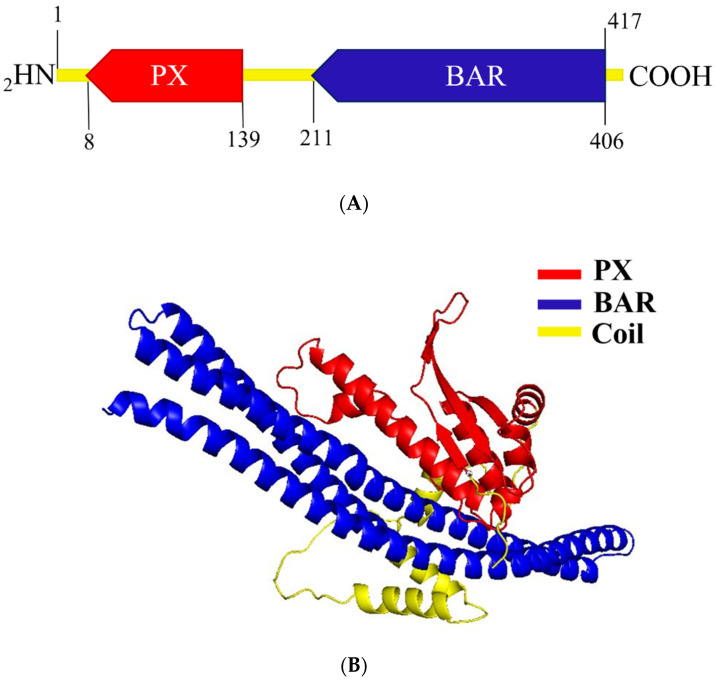
Domain organization and tertiary structure prediction of A0A504X805. (**A**) Schematic representation of the A0A504X805 domain architecture. The N′ and C′ terminal aa residues are indicated above the schematic diagram while the beginning and end of the PX and BAR domains are indicated below. (**B**) Tertiary structure prediction of *Ld*SNXi with PyMol tool based on data retrieved from Alpha Fold. Red depicts the PX domain, blue depicts the BAR domain, and yellow shows the coil region.

**Figure 3 ijms-25-04095-f003:**
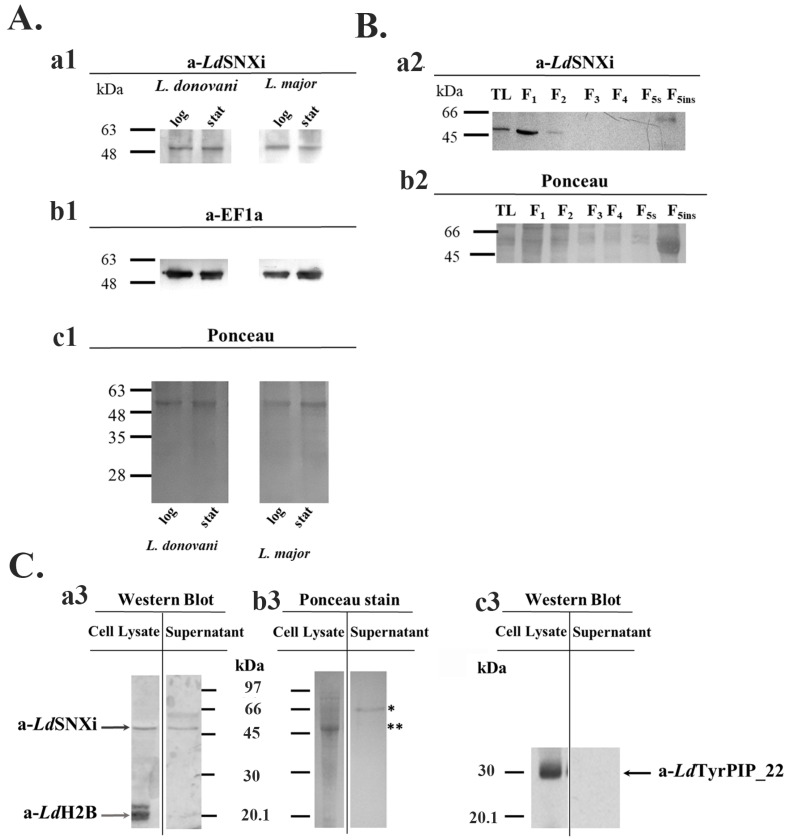
Biochemical detection of *Ld*SNXi in cultured *Leishmania* cells and in the extracellular medium. (**A**) Detection of the endogenous SNXi in wild type *L. donovani* (LG13) and *L. major* (Friedlin) promastigotes. Total lysates (40 μg) (Section 4) were analyzed by SDS-PAGE and Western blot probed with the a-*Ld*SNXi pAb (0.5 μg/mL) (**a1**). The same membrane, after stripping, was re-blotted with the a-EF1a mAb (1:10,000) (**b1**). *Ld*EF1a was used as a loading indicator. The membrane stained with Ponceau-S is shown (**c1**) as a second loading indicator. (**B**) Subcellular distribution of *Ld*SNXi in *wt L. donovani* (LG13) fractions generated by stepwise solubilization of stationary phase promastigote pellets with digitonin and Triton X-100 (Section 4). Western blot with the rabbit a-*Ld*SNXi pAb (**a2**). Ponceau-S of respective membrane regions is shown in (**b2**) as loading indicator. (**C**) Biochemical detection of the endogenous *Ld*SNXi in *wt L. donovani* (LG13) promastigotes and in the extracellular material of the promastigote culture (secreted fraction). (**a3**) Western blot with the purified rabbit a-*Ld*SNXi pAb (0.5 μg/mL) and rabbit a-*Ld*Histone2B (H2B) serum (1:1000) used together. (**c3**) Western blot with the a-*Ld*TyrPIP_22 mouse pAb (1:2500) after stripping of the Abs used in (**a3**). The membrane stained with Ponceau-S is shown in (**b3**) as loading indicators. The black arrows point to the bands identified by the a-*Ld*SNXi and a-*Ld*TyrPIP_22 specific pAbs at the expected migration position, while the gray arrow points to the band identified by the a-*Ld*H2B specific rabbit pAb. Molecular weights are indicated in kDa in the middle of the two panels. The asterisk indicates the migration position where the major secreted protein GP63 is expected, while the two asterisks indicate the major cytoskeleton protein tubulin. Molecular weights are indicated in kDa on the left of the panels. Different molecular weight markers were used in A–C (Section 4). TL: total lysate. F1–F5ins: the detergent fractionation fractions.

**Figure 4 ijms-25-04095-f004:**
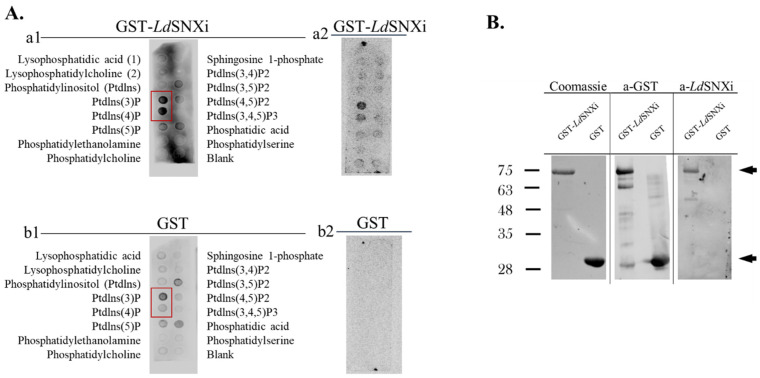
PI binding specificity of GST-*Ld*SNXi. (**A**) Recombinant GST-*Ld*SNXi (**a1**,**a2**) and GST (**b1**,**b2**) were incubated with nitrocellulose membranes (PIP strips) spotted with phosphoinositides and other lipids, as described in Section 4. In (**a1**,**b1**), equal ng/mL protein were used (i.e., 50 ng/mL). In (**a2**,**b2**) were used equimolar protein concentrations (i.e., 50 ng/mL for GST-*Ld*SNXi and 17 ng/mL for GST respectively). The signal for PtdIns(3)*P* and PtdIns(4)*P* is framed by a red rectangle in (**a1**,**b1**). (**B**) Purified GST-*Ld*SNXi and GST from lysates of the corresponding *E. coli* BL21 recombinant clones used in the PI binding assay were analyzed by SDS-PAGE (Coomassie) and Western blot (a-GST and a-*Ld*SNXi). The arrows on the right indicate the protein species with the expected size for GST*-Ld*SNXi (upper arrow) and GST (lower arrow), respectively.

**Figure 5 ijms-25-04095-f005:**
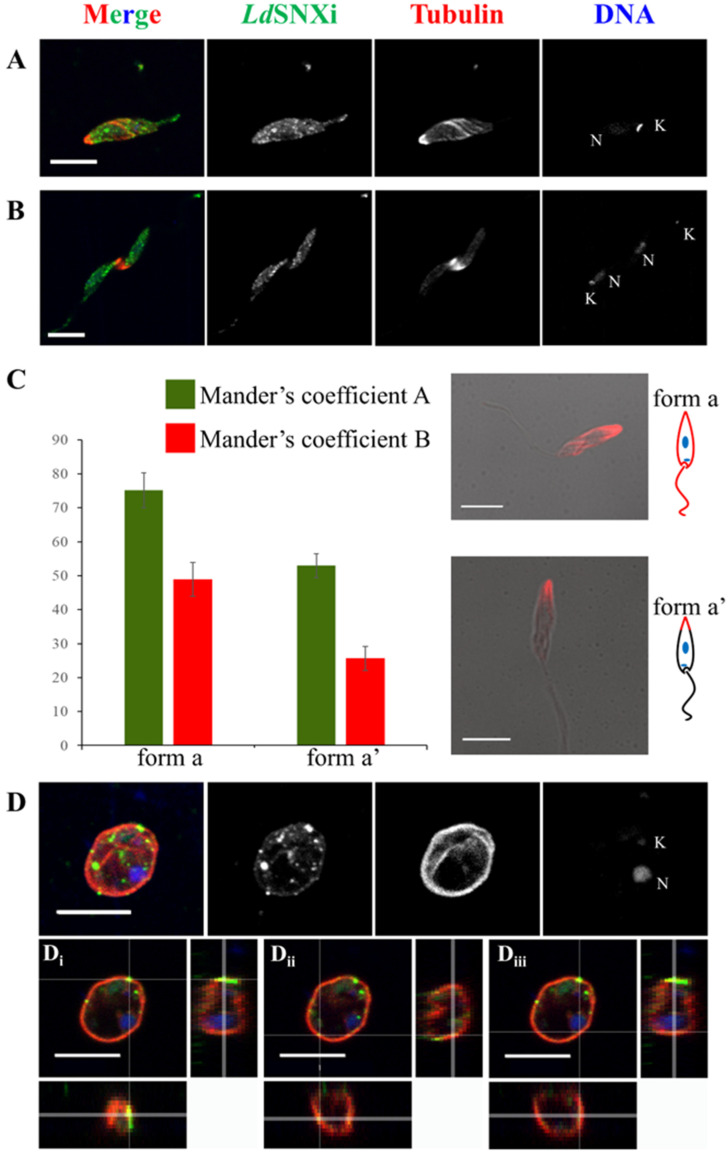
Localization of endogenous *Ld*SNXi in *L. donovani* promastigotes and axenic amastigotes. Localization of *Ld*SNXi in different cell cycle-dependent morphological forms of cultured *L. donovani* promastigotes; co-staining for *Ld*Tubulin. Maximum intensity projection images from each case are shown. Single FL images are shown in black and white (BW) for better contrast while images of the merged FL signals are shown in color. The molecule highlighted in the BW images with single-color FL is indicated at the top in the same color as the respective FL signal. (**A**) Procyclic-like promastigote. (**Β**) Dividing promastigote with the intercellular bridge highlighted by the tubulin staining. (**C**) Mander’s coefficient graphs depicting the percentage of pixels in the co-localization channel, as calculated by IMARIS (Section 4). Two different forms of promastigotes depending on the observed microtubules’ staining pattern were grouped together. Merged images of phase contrast and FL images of microtubule staining (Red FL) in the two representative parasite forms are shown on the right, together with schematic representations of the microtubule staining distribution. (**D**) Amastigote-like parasite. (**i**,**ii**) are orthogonal images of the same parasite showing the position of different *Ld*SNXi stained vesicles in relation to the parasite’s surface membrane. (**iii**) is an orthogonal image of the same parasite showing the co-localization of *Ld*SNXi epitopes and subpellicular microtubules. Red FL: Tubulin. Green FL: *Ld*SNXi. Κ: kinetoplast DNA. N: nuclear DNA. Scale bar size: 5 μm.

**Figure 6 ijms-25-04095-f006:**
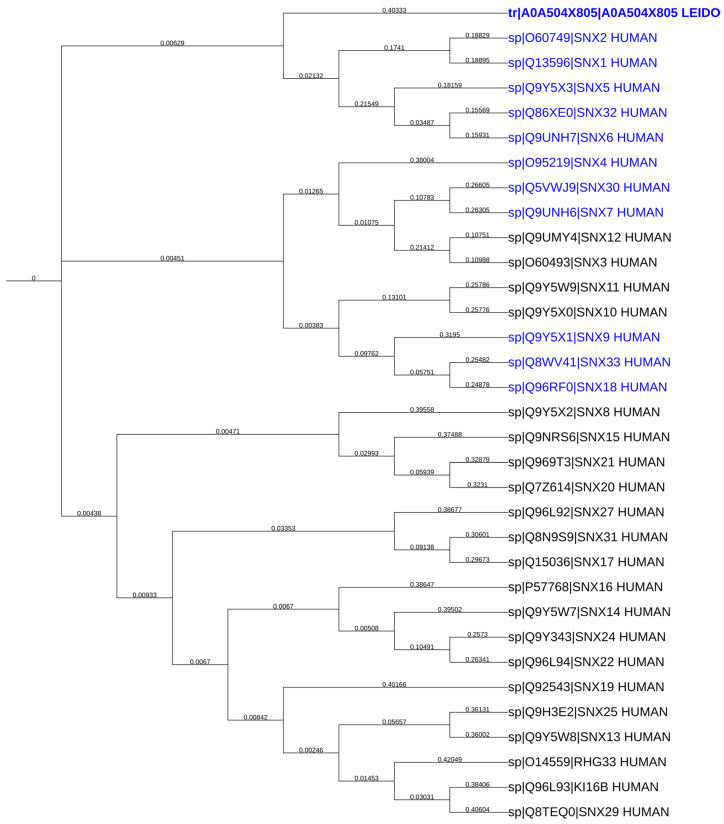
Evolutionary relationship between the *Ld*BPK 352470.1 gene product (A0A504X805) and the human sorting nexins. Comprehensive neighbor-joining phylogenetic trees generated using iTOL based on multiple sequence alignment (ClustalW) of the LdBPK 352470.1 gene product and all reviewed human sorting nexins (retrieved from UniProt) https://itol.embl.de/tree/14123716952488761694711835 accessed on 10 June 2023. The members of the human SNX-BAR subfamily (according to UniProt) are shown in blue. Branch lengths are displayed on each corresponding branch of the phylogram. Statistically significant % identities of the sequences are shown in Table 2. *Ld*SNXi is depicted in bold.

**Figure 7 ijms-25-04095-f007:**
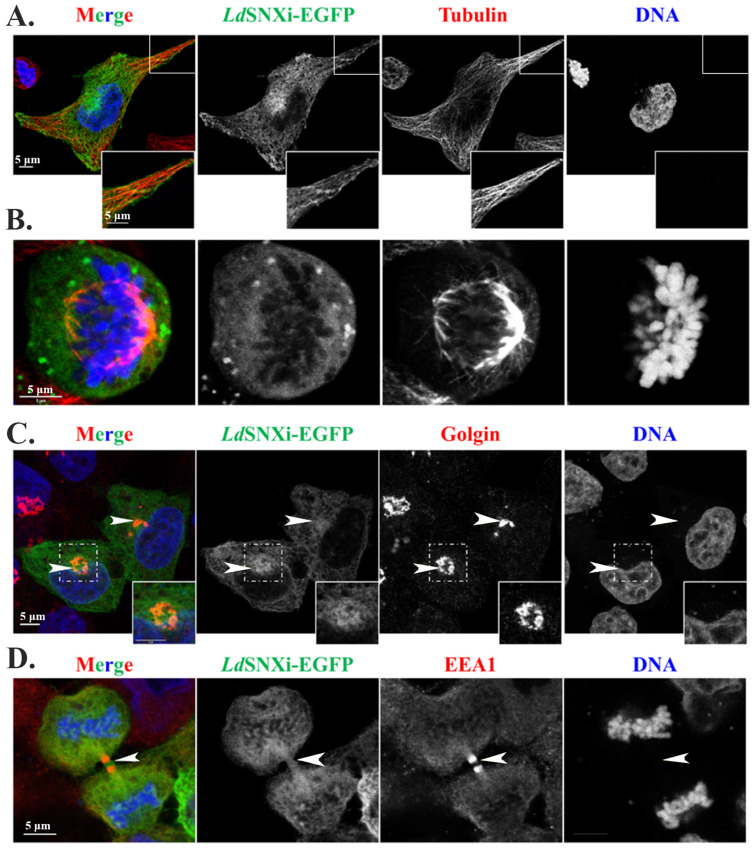
Localization of r*Ld*SNXi-EGFP heterologously expressed in human HeLa cells. HeLa cells were transfected with the pEGFP-N3-*ldsnxi* plasmid for 24 h, followed by pre-extraction and immunostaining, as described in Section 4. (**A**). An Interphase cell. (**B**). A metaphase cell. (**C**). Interphase cells stained for Golgi. (**D**). A cell in late cytokinesis stained for the early endosome protein EEA1. Insets indicate the area framed in the main figure in 2× zoom. Arrows point to the Golgi apparatus and the intercellular bridge. Single FL images are shown in BW for better contrast, while images of the merged FL signals are shown in color. The molecule highlighted in the BW images with single color FL is indicated at the top in the same color as the respective FL signal. Scale bar size: 5 μm.

**Figure 8 ijms-25-04095-f008:**
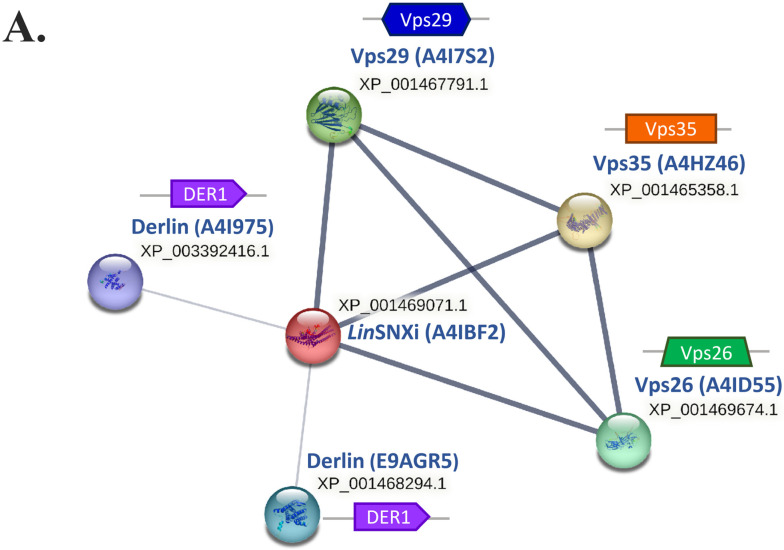
Molecular partners of *Ld*SNXi and human SNXs according to STRING database analysis. (**A**) Protein–protein interactions network of *Lin*SNXi (ortholog protein of the *Ld*SNXi in *Leishmania infantum*) and its partners according to STRING analysis based on experimental evidence. The characteristic domain of each protein (according to Pfam) is depicted next to each node of the network. (**B**) Comparative illustration for the characteristic domains of the molecular partners of *Lin*SNXi and human SNX1, SNX2, SNX4, and SNX7.

**Table 1 ijms-25-04095-t001:** Blastp pairwise sequence alignment of *Ld*SNXi (LdBK 352470.1 gene product) and Trypanosomatidae orthologs from TriTrypDB. Herein are shown the statistically significant matches calculated by the algorithm using default parameters. The sequences are listed in the same order that they appear in the phylogram in Figure 1.

Protein ID	Query Cover	E-Value	Per. Ident.	Acc. Len
LARLEM1108_350030500 (*Leishmania arabica*)	100%	0.0	97.36%	417
LmjF.35.2420 (*Leishmania major*)	100%	0.0	97.60%	417
LGELEM452_350030200 (*Leishmania gerbilli*)	100%	0.0	97.60%	417
LMJLV39_350031000 (*Leishmania major*)	100%	0.0	97.36%	417
LTULEM423_350030400 (*Leishmania turanica*)	100%	0.0	97.12%	417
LTRL590_350029600 (*Leishmania tropica*)	100%	0.0	97.84%	417
LAEL147_000774600 (*Leishmania aethiopica*)	100%	0.0	97.12%	417
LdCL_350029600 (*Leishmania donovani*)	100%	0.0	99.76%	417
LINF_350029500 (*Leishmania infantum*)	100%	0.0	99.76%	417
LDHU3_35.3170 (*Leishmania donovani*)	100%	0.0	100.00%	417
LmxM.34.2420 (*Leishmania mexicana*)	99%	0.0	97.12%	418
LAMA_000790800 (*Leishmania amazonensis*)	99%	0.0	96.40%	418
LtaP35.2450 (*Leishmania tarentoloe*)	100%	0.0	92.11%	418
LtaPh_3524500 (*Leishmania tarentoloe*)	100%	0.0	92.11%	418
LbrM.34.2.002370 (*Leishmania braziliensis*)	100%	0.0	92.34%	418
LbrM.34.2370 (*Leishmania braziliensis*)	100%	0.0	92.34%	418
LPAL13_340028800 (*Leishmania panamensis*)	100%	0.0	92.82%	418
LPMP_342280 (*Leishmania panamensis*)	100%	0.0	92.82%	418
LBRM2903_340031200 (*Leishmania braziliensis*)	100%	0.0	92.82%	418
LENLEM3045_350030200 (*Leishmania enrietti*)	100%	0.0	90.91%	418
Lsey_0008_0590 (*Leptomonas seymouri*)	99%	0.0	77.18%	426
LpyrH10_01_4540 (*Leptomonas pyrrhocoris*)	99%	0.0	75.59%	427
TM35_000044480 (*Trypanosoma theileri*)	98%	1 × 10^−68^	33.88%	415
DQ04_03391000 (*Trypanosoma grayi*)	96%	1 × 10^−71^	34.13%	426
TcIL3000.A.H_000702400 (*Trypanosoma congolense*)	98%	8 × 10^−58^	31.24%	419
TevSTIB805.9.9680 (*Trypanosoma evansi*)	97%	9 × 10^−67^	31.24%	419
Tbg972.9.8370 (*Trypanosoma brucei gambiense*)	97%	9 × 10^−67^	31.24%	419
Tb927.9.13380 (*Trypanosoma brucei brucei*)	97%	9 × 10−^67^	31.24%	419
Tb427_090080200 (*Trypanosoma brucei lister*)	97%	9 × 10^−67^	31.24%	419
Tb427tmp.211.4240 (*Trypanosoma brucei lister*)	97%	7 × 10^−66^	31.00%	419
TRSC58_00327 (*Trypanosoma rangeli*)	98%	1 × 10^−67^	33.18%	422
Tc_MARK_4526 (*Trypanosoma cruzi marinkellei strain B7*)	97%	1 × 10^−70^	33.25%	422
C3747_71g21 (*Trypanosoma cruzi TCC*)	97%	4 × 10^−73^	34.43%	422
TcCL_NonESM00301 (*Trypanosoma cruzi CL*)	97%	4 × 10^−73^	34.43%	429
TcCLB.510657.30 (*Trypanosoma cruzi CL Brener Non-Esmeraldo*)	97%	4 × 10^−73^	34.43%	422
TcCL_ESM00146 (*Trypanosoma cruzi CL*)	97%	2 × 10^−74^	34.43%	422
TcYC6_0068490 (*Trypanosoma cruzi Y C6*)	97%	2 × 10^−74^	34.43%	422
C3747_1g93 (*Trypanosoma cruzi TCC*)	97%	2 × 10^−74^	34.43%	422
TcCLB.510749.30 (*Trypanosoma cruzi CL Brener Esmeraldo*)	97%	2 × 10^−74^	34.43%	422
TcG_02026 (*Trypanosoma cruzi G*)	97%	8 × 10^−75^	34.67%	422
TCSYLVIO_001384 (*Trypanosoma cruzi sylvio*)	97%	8 × 10^−75^	34.67%	422
C4B63_2g85 (*Trypanosoma cruzi Dm28*)	97%	8 × 10^−75^	34.67%	422
BCY84_13957 (*Trypanosoma cruzi Dm28*)	97%	8 × 10^−75^	34.67%	422
TCDM_03350 (*Trypanosoma cruzi Dm28*)	97%	8 × 10^−75^	34.67%	422
TcBrA4_0062990 (*Trypanosoma cruzi*)	97%	8 × 10^−75^	34.67%	422

**Table 2 ijms-25-04095-t002:** Blastp pairwise sequence alignment of *Ld*SNXi (LdBK 352470 gene product) and human SNX-BAR subfamily members of sorting nexins. Shown are only the statistically significant matches calculated by the algorithm using a threshold of 0.05 and default parameters. The proteins are listed as their corresponding sequences in the phylogram presented in Figure 6.

Protein ID	Query Cover	E-Value	Per. Ident.	Acc. Length
SNX2	96%	2 × 10^−19^	23.56%	519
SNX1	96%	4 × 10^−20^	24.11%	522
SNX5	28%	2 × 10^−6^	28.03%	404
SNX32	13%	1 × 10^−2^	33.85%	403
SNX4	23%	1 × 10^−1^	36.00%	450
SNX7	30%	3 × 10^−11^	32.31%	387
SNX3	31%	1 × 10^−8^	26.72%	162
SNX9	23%	9 × 10^−69^	23.76%	595
SNX33	31%	2 × 10^−62^	24.44%	574
SNX18	31%	4 × 10^−94^	26.43%	628

**Table 3 ijms-25-04095-t003:** Blastp pairwise sequence alignment of the individual PX (top) and BAR (bottom) domains of *Ld*SNXi (*Ld*BK 352470.1 gene product) and human SNX-BAR sorting nexins subfamily members. Shown are only the statistically significant matches calculated by the algorithm using default values.

	Human SNX	Query Cover	E-Value	Per. Ident.
**PX**				
	PX_SNX4 (61-187)	68%	4 × 10^−15^	38.30%
	PX_SNX7 (30-151)	97%	8 × 10^−14^	32.31%
	PX_SNX2 (140-269)	84%	2 × 10^−13^	31.67%
	PX_SNX30 (89-210)	93%	6 × 10^−11^	26.02%
	PX_SNX1 (143-272)	84%	1 × 10^−10^	30.83%
	PX_SNX18 (276-386)	91%	8 × 10^−10^	26.83%
	PX_SNX8 (73-181)	65%	5 × 10^−9^	28.74%
	PX_SNX5 (25-172)	88%	1 × 10^−8^	28.03%
	PX_SNX9 (250-361)	66%	2 × 10^−6^	23.60%
	PX_SNX33 (230-340)	90%	2 × 10^−6^	24.79%
	PX_SNX32 (20-168)	41%	2 × 10^−4^	34.38%
	PX_SNX6 (26-173)	46%	5 × 10^−3^	30.99%
**BAR**				
	BAR_SNX2_human (299-519)	91%	2 × 10^−8^	21.72%

## Data Availability

Data are contained within the article and Appendix A.

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
