# Peer review of "Characterization of the First Secreted Sorting Nexin Identified in the Leishmania Protists"

_ijms, 2024, doi:10.3390/ijms25074095_

Round 1
Reviewer 1 Report
Comments and Suggestions for Authors
Comments on the Quality of English LanguageRequire minor editing
Author Response
We thank Reviewer 1 for the positive evaluation of our manuscript and for all the comments highlighting experiments that would strengthen our conclusions and complement the study presented in the manuscript under reviewing. His/Her comments are all very valuable and certainly some of them are the focus of ongoing and future work in our lab. Currently we are applying for funding and seeking to establish crucial collaborations in order to continue this work along the lines the reviewer is suggesting. Some of the reviewer’s comments are addressed in the conclusions section. See phrases highlighted in cyan.

Reviewer 2 Report
Comments and Suggestions for Authors
- A brief summary
The manuscript reported the evolutionary conservation of SNX- BAR subfamily οf SNXs. LdSNXi, the novel SNX characterized in this study, is a molecule that hijacks functions of host cells’ SNXs. The study reported also Blast sequence alignment, phylogenetic analysis, 2D and 3D structure predictions.
· General concept comments
The manuscript is relevant for the field and presented in a well-structured way. Experimental design is appropriate, and results are reproducible. However, there are some typing/formatting errors that should be fixed. All the abbreviations in the text should be explained in their first appearance. Some of them are not clear in the abstract. The cited references are not reported in the correct form. The conclusions highlighted the work limitations, but they are not consistent with the arguments presented. I suggest reorganizing the conclusions section. Data availability statement has not been found.
- Specific comments
Line 38: There is an error in the keywords. Please, correct it.
Line 117: “A0A504X805,” is not clear. Is it the Uniprot code?
Comments on the Quality of English LanguageMinor editing of English language required
Author Response
We thank the reviewer for the positive comments concerning the presentation structure of our manuscript, its contribution to the field, the experimental design and the reproducibility of the results.
Replies to general concept comments
- Typing and formatting errors were corrected as much as possible. If there are any which have escaped our attention, please let us know.
- Corrections were introduced concerning the abbreviations. They are now all explained the first time they appear in the text. Please see highlighted text in Abstract, Introduction and Materials and Methods sections.
- Two new references were cited highlighted in green. The numbers of the references have been changed accordingly. Could the reviewer indicate which references he/she finds not appropriately cited? We will try to change them.
4) The conclusions’ section was significantly reorganized/rewritten.
Replies to Specific comments
Line 38: The error was corrected.
Line 117: Indeed. The A0A504X805 is a UniProtKB code. A correction was introduced making this point clear. See text highlighted in yellow

Reviewer 3 Report
Comments and Suggestions for Authors
The manuscript is quite interesting, and the work done by the authors is also a significant contribution to understanding the pathogenesis of Leishmania by characterizing its first SNX protein. Below are a few minor queries and comments:
Title: No need to include "Kinetoplastea protist" in the title.
Keywords: What does "7" represent in the keywords?
The abstract and the introduction's first paragraph are almost same. It is advisable to modify either the abstract or the introduction's start. The introduction should commence with "Leishmania donovani."
Line 87: The first line of the abstract states that SNXs are a large group of eukaryotic proteins; however, bacterial examples are given in line 87. Please revise the abstract statement if you intend to include bacterial examples for SNX.
It is not necessary to include all results in the main text; some unnecessary results should be placed in the supplementary information.
Table 1: It should not be included in the main manuscript; it is merely a blast analysis. For example, Figure 1 is suitable for inclusion in the main manuscript.
Figure 1: The iTOL link and parameters used should be included in the methods. The phylogram should include individual branch lengths as well as the percentage sequence identity at the end of each protein compared to your LdBKP 352470.1 protein (100%).
Figures 1 and 2 can be merged. The phylogram can be shortened by including 20-30 key similar sequences.
Figure 3: SDS-PAGE should not be accepted without a protein marker/ladder loaded with samples, whereas Western blot can be accepted, provided full blot pictures are given in the supplementary information. This can be any protein, and you have only labelled the reference molecular weight; it is challenging for a reviewer to believe how you express and purify your protein of interest.
Figure 4B is also without real markers.
Figure 5: What do "K" and "N" represent? This information should be mentioned in the figure legend. Why are pixels visible in Figure 5D? Have the pictures been cropped and zoomed? This should not be done. The scales are also not labelled.
Table 2: Sequence ID is not mentioned.
Figure 6: It shows that LdSNXi is significantly evolutionarily distant with lower identity from human SNXs, whereas in some cases, coverage is also very low. Please provide justification. Branch length and % similarity or identity should be given for each sequence in the phylogram.
Why are HeLa cells used for the transfection of rLdSNXi? However, Leishmania infects macrophages and neutrophils naturally, so the human monocytic THP-1 cell line should be used.
Figure 7: The legend is not detailed, and the micrographs are without a labelled scale.
Figure 8: is very useful for interpreting the structural difference between LdSNXi and human SNX.
Comments on the Quality of English LanguageMinor editing of the English language required
Author Response
Minor queries and comments
Reviewer: Title: No need to include "Kinetoplastea protist" in the title.
Reply: The title was modified as: “Characterization of the First Secreted Sorting Nexin identified in the Leishmania protists”
Reviewer: Keywords: What does "7" represent in the keywords?
Reply: The number 7 was removed from keywords. It was an error
Reviewer: The abstract and the introduction's first paragraph are almost same. It is advisable to modify either the abstract or the introduction's start. The introduction should commence with "Leishmania donovani."
Reply: The abstract’s first paragraph was modified (highlighted in cyan). Given that the topic of the manuscript is a sorting nexin, we were prompted to start with it and the related background knowledge. Moreover, making the Introduction’s start with "Leishmania donovani" this would require major revisions of the Introduction, and perhaps this change would raise significant comments/objections from the other two reviewers.
Reviewer: Line 87. The first line of the abstract states that SNXs are a large group of eukaryotic proteins; however, bacterial examples are given in line 87. Please revise the abstract statement if you intend to include bacterial examples for SNX.
Reply: SNXs are eukaryotic proteins. The examples given at line 87 are for bacteria proteins that highjack the function of eukaryotic SNXs. These bacteria proteins are not SNXs themselves. A correction was introduced (line 85, text highlighted in green) to make this point clearer.
Reviewer: It is not necessary to include all results in the main text; some unnecessary results should be placed in the supplementary information. Table 1: It should not be included in the main manuscript; it is merely a blast analysis. For example, Figure 1 is suitable for inclusion in the main manuscript.
Reply:
We appreciate the attention of the reviewer to details of our work, as well as the suggestion regarding excluding some information from the main manuscript. While Table 1 does indeed present detailed blast analysis results, it serves a significant informational purpose within the context of our study. In addition, the next comment asks for this precise information to be given within the figure of the phylogenetic tree in Figure 1. Such an illustration would be nice, and we indeed have tried it. However, it would be very difficult for the reader to discern what it was written. Since the information about % identity and query cover is of great importance, we created a new Table 1 where the proteins are presented in the same order as in the phylogram of Figure 1, making it thus easier for the reader of the manuscript to identify these information pieces for each sequence listed in the phylogram.
Reviewer: Figure 1: The iTOL link and parameters used should be included in the methods. The phylogram should include individual branch lengths as well as the percentage sequence identity at the end of each protein compared to your LdBKP 352470.1 protein (100%).
Reply:
Figure 1 was modified to include the individual branch length on each corresponding branch. The parameters used in this analysis were the Default scaling factors despite the vertical and horizontal scaling which were adjusted to 1.2. This information was added in the corresponding figure legend.
The iTOL link: https://itol.embl.de/tree/1939213722931771674054468 was also added in the Figure legend. As for the Percentage identity, it was not added on the Figure 1 for reasons of illustration and reasons explained in the previous reply. The Percentage identity is presented in Table 1, in which the protein sequences appear in the same order as in the phylogram of Figure 1.
Reviewer: Figures 1 and 2 can be merged. The phylogram can be shortened by including 20-30 key similar sequences.
Reply:
Figure 1 is indeed a large Figure. The phylogenetic tree has been constructed after systematic retrieval of homologous proteins complying with certain parameters/thresholds as reported in material section and as requested, in the previous comment. The figure is not meant to give few representatives of the homologous, but rather all homologues emphasizing the robustness of our search and analysis. Moreover, this way, “similar” sequences are defined with exact criteria. For this reason, we left all retrieved proteins within Figure 1.
If the Reviewer does not consider absolutely essential to merge Figures 1 and 2 we would like to keep Figure 2 and keep chapters 3.1 and 3.2 separate as they deal with different aspects of the bioinformatics analysis.
Reviewer: Figure 3: SDS-PAGE should not be accepted without a protein marker/ladder loaded with samples, whereas Western blot can be accepted, provided full blot pictures are given in the supplementary information. This can be any protein, and you have only labelled the reference molecular weight; it is challenging for a reviewer to believe how you express and purify your protein of interest. Figure 4B is also without real markers.
Reply: Figures 3 and 4 are images of the same Ponceau-S Stained membrane used for the Western Blot presented in the same Figures. They are not images of SDS-PAGE. These Figures do not correspond to expression and purification experiments of the proteins of interest. In these experiments have been analyzed total lysates of Leishmania cells or culture supernatants from Leishmania promastigote culture. Originals for all the images in Figures 3 and 4 have been provided to the IJMS editorial office. We could provide them again if the reviewer requests them.
Reviewer: Figure 5: What do "K" and "N" represent? This information should be mentioned in the figure legend.
Reply: “K” represents Kinetoplast DNA and “N” represents Nuclear DNA. A line was added at the end of the Figure 5 legend to clarify this point (highlighted in yellow).
Reviewer: Figure 5: Why are pixels visible in Figure 5D? Have the pictures been cropped and zoomed? This should not be done.
Reply: The image was acquired in the appropriate format (63x objective, zoom 4, 792x792 pixels, thus each pixel contains the information corresponding to the smallest area optically possible. Digital zooming during image acquisition is making each pixel bigger, but it does not alter the original information. Original images can be provided if the reviewer requests.
Reviewer: Figure 5. The scales are also not labelled.
Reply: All scale bars represent size of 5μm, as stated in all Figure legends from figures with images.
Reviewer: Why are HeLa cells used for the transfection of rLdSNXi? However, Leishmania infects macrophages and neutrophils naturally, so the human monocytic THP-1 cell line should be used.
Reply: We agree with the reviewer. However, we encountered problems in efficiently transfecting THP-1 cells. Moreover, the size and shape of HeLa cells is more appropriate to evaluate detailed intracellular localization of proteins by confocal microscopy.
Reviewer: Figure 6: It shows that LdSNXi is significantly evolutionarily distant with lower identity from human SNXs, whereas in some cases, coverage is also very low. Please provide justification. Branch length and % similarity or identity should be given for each sequence in the phylogram.
Reply:
The evolutionary divergence and reduced identity observed between LdSNXi and human SNXs is expected given the extensive evolutionary timescale distance of the human and Leishmania organisms. Usually protein structures exhibit greater conservation as compared to their sequences across prolonged evolutionary periods. Hence, in numerous cases, the identification of orthologous proteins relies on domain conservation rather than sequence similarity/identity.
Figure 6 was modified. Branch lengths were added on the corresponding branches.
Reviewer: Figure 7: The legend is not detailed, and the micrographs are without a labelled scale.
Reply: A mistake was made in the formatting of the manuscript. The legend was there but appeared as part of the Results section text. Correction was introduced. All scale bars represent size of 5μm, as stated in the figure legend.
Reviewer: Figure 8: is very useful for interpreting the structural difference between LdSNXi and human SNX.
Reply: We considered this figure necessary for the visual representation of these data. We are glad it was successful. A small correction was made on the Figure. The word FYVE appeared as FYEV in the previous figure.
